# The Prevalence of Carbapenemase-Producing Microorganisms and Use of Novel Cephalosporins for the Treatment of Severe Infections Caused by Carbapenem-Resistant Gram-Negative Bacteria in a Pediatric Cardiac Intensive Care Unit

**DOI:** 10.3390/antibiotics12050796

**Published:** 2023-04-22

**Authors:** Costanza Tripiciano, Lorenza Romani, Stefania Mercadante, Laura Cursi, Martina Di Giuseppe, Francesca Ippolita Calo Carducci, Tiziana Fragasso, Luca Di Chiara, Cristiana Garisto, Annamaria Sisto, Leonardo Vallesi, Valentino Costabile, Laura Lancella, Paola Bernaschi, Maia De Luca

**Affiliations:** 1Infectious Disease Unit, Bambino Gesù Children’s Hospital, IRCCS, 00165 Rome, Italy; 2Pediatric Cardiac Intensive Care Unit, Cardiology and Cardiac Surgery, Bambino Gesù Children’s Hospital, IRCCS, 00165 Rome, Italy; 3Unit of Microbiology and Diagnostic Immunology, Bambino Gesù Children’s Hospital, IRCCS, 00165 Rome, Italy; 4Hospital Pharmacy Unit, Bambino Gesù Children’s Hospital, IRCCS, 00165 Rome, Italy

**Keywords:** carbapenemase, novel cephalosporins, pediatric ICU

## Abstract

Background: The spread of carbapenem-resistant organisms (CROs) is an increasingly serious threat globally, especially in vulnerable populations, such as intensive care unit (ICU) patients. Currently, the antibiotic options for CROs are very limited, particularly in pediatric settings. We describe a cohort of pediatric patients affected by CRO infections, highlighting the important changes in carbapenemase production in recent years and comparing the treatment with novel cephalosporins (N-CEFs) to Colistin-based regimens (COLI). Methods: All patients admitted to the cardiac ICU of the Bambino Gesù Children’s Hospital in Rome during the 2016–2022 period with an invasive infection caused by a CRO were enrolled. Results: The data were collected from 42 patients. The most frequently detected pathogens were *Pseudomonas aeruginosa* (64%), *Klebsiella pneumoniae* (14%) and *Enterobacter spp.* (14%). Thirty-three percent of the isolated microorganisms were carbapenemase producers, with a majority of VIM (71%), followed by KPC (22%) and OXA-48 (7%). A total of 67% of patients in the N-CEF group and 29% of patients in the comparative group achieved clinical remission (*p* = 0.04). Conclusion: The increase over the years of MBL-producing pathogens in our hospital is challenging in terms of therapeutic options. According to the present study, N-CEFs are a safe and effective option in pediatric patients affected by CRO infections.

## 1. Introduction

The problem of antimicrobial resistance (AMR) has increased significantly in recent years, considerably affecting public health. The constant, and too often inappropriate, use of antibiotics increases selective pressures, encouraging the emergence, multiplication and spread of resistant strains. Estimates of AMR-associated mortality are alarming, as reported by the O’Neill’s 2014 survey, which predicted that there will be around 10 million deaths per year due to antibiotic resistance by 2050 [1]. Over the past decades, international organizations such as the World Health Organization (WHO) and the European Centre for Disease Prevention and Control (ECDC) have produced recommendations and proposed coordinated strategies to contain the phenomenon, recognizing AMR as a health priority [2]. AMR is often responsible for healthcare-related infections that arise and spread within hospitals or other healthcare facilities. In particular, “carbapenem-resistant organism” (CRO) infections are one of the main enemies in terms of global public health due to the high mortality caused by the delayed administration of appropriate therapy and limited availability of treatment options. There is often no effective treatment for infected patients and, consequently, the spread of carbapenemase-producing bacteria could be the vanishing point for AMR-related morbidity and mortality [1,2,3]. Risk factors for the infections caused by CROs include prolonged exposure to antibiotics, comorbidities, invasive procedures, and the use of medical devices and mechanical ventilation [4]. Therefore, carbapenem-resistant bacteria infections are more common in Intensive Care Units (ICUs) than in other inpatient wards or in outpatients [5], with important consequences in terms of mortality, length of hospitalization and economic costs [6]. Patients admitted to cardiac intensive care units (CICU) have specific risk factors that expose them to infections even more than in other ICUs, such as the use of several vascular devices, the immunosuppressive treatment for heart transplant or the immunoparalysis due to cardiopulmonary bypass circuitry or ischemia-reperfusion injury [7]. A variable proportion of 2–20% of the patients undergoing open-heart surgery suffers from infections in the postoperative period, such as ventilator associated pneumonia (VAP), surgical site infections and bloodstream infections [8].

The choice of an antimicrobial regimen for carbapenem-resistant Gram-negative infections is often challenging, although the level of difficulty depends on the specific clinical scenario. Moreover, susceptibility patterns are not predictable for most carbapenem-resistant Gram-negative bacteria, so the selection of the antibiotic must be guided by antibiotic-specific susceptibility test results [9]. At present, antibiotic options for the treatment of carbapenem-resistant bacteria are very limited and historically included polymyxins, tigecycline, and aminoglycosides as cornerstones. Due to the lack of alternative therapeutic regimes, there is an increasing need for new and effective therapies for infections due to bacteria that are not susceptible to carbapenems. In recent years, new therapeutic strategies have been studied and identified, including high-dose tigecycline, high-dose prolonged infusion of carbapenems and dual-carbapenem therapy. Recently, potentially promising new antibiotics have become available such as ceftazidime/avibactam, which is active against KPC and OXA-48 producers; ceftolozane/tazobactam, which is particularly active against difficult-to-treat *Pseudomonas*; meropenem/vaborbactam, which is active against KPC producers; plazomycin, which is a new-generation aminoglycoside with in vitro activity against carbapenem-resistant *Enterobacterales* (CRE); and eravacyclin. However, for most of them, data on their safety and efficacy in the pediatric population are limited [10]. Hopefully, these new agents could be incorporated into future carbapenem-sparing strategies for the treatment of multi-drug-resistant (MDR) bacteria, helping to contain the rapid spread of carbapenemase-producing bacteria [10].

Against this background, we conducted a retrospective study to describe a cohort of patients admitted to the pediatric ICU of our hospital with severe infections caused by CROs, in order to evaluate the trend of the CROs infections and carbapenemase production in the last few years in our hospital. The secondary aim of the study was to compare the outcomes in patients treated with new-generation cephalosporins (N-CEFs) to patients treated with colistin-containing regimens (COLI).

## 2. Results

Among all patients admitted to the cardiac ICU (CICU) of the Bambino Gesù Children’s Hospital between 2016 and 2022, a Gram-negative carbapenem-resistant organism was detected in a sterile site (e.g., blood, urine or bronchoalveolar fluid) in 58 patients. Among these patients, sixteen were excluded due to the lack of signs or symptoms of infection, or because the isolated microorganism was not held responsible for the current infection. The final cohort consisted of 42 patients, 57% of whom were male, with a mean age of 9.7 years. Based on the admission cause, patients were divided into three categories: post-surgical patients (e.g., valve replacement surgery); patients undergoing cardiac, pulmonary or cardiopulmonary transplant; and patients with heart failure requiring intensive care. Approximately 60% of the patients had an associated comorbidity, of which immunodeficiency accounted for 50%, nephropathy for 3% and genetic syndromes for 27%; 11% of the cases included miscellaneous conditions such as cystic fibrosis or anatomical malformations. Moreover, 15% of the patients were premature (gestational age < 37 weeks).

In 16% of the cases the patients were admitted from the Emergency Room, in 57% of the cases they had been transferred to the ICU from another inpatient ward and in 26% of the cases from another intensive care unit.

Fifty-two percent of the patients were colonized by a carbapenem-resistant pathogen before developing the infection. Among these, 62% of the pathogens isolated were carbapenemase-producers, with a clear predominance of VIM (73%). *Pseudomonas aeruginosa* was the most frequent colonizer (46%), followed by *Enterobacter cloacae* (33%), *Klebsiella pneumoniae* (17%) and *Acinetobacter baumanii* (4%). In 91% of the cases, a match between the species of colonizing microorganism and the etiological agent of the infection was found. However, in 10% of these cases, the production of carbapenemase of the colonizing CRO was not confirmed in the pathogen associated with the infection.

Almost all infections were nosocomial (95%), with a time between hospital admission and infection of up to 86 days, and between admission to CICU and development of the infection of up to 70 days.

The most frequent clinical presentation was pneumonia (64%), followed by sepsis/septic shock (34%) and urinary tract infection (7%). In addition, two patients had soft tissue infections, while one patient developed necrotizing enterocolitis. The most frequently isolated pathogens were *Pseudomonas aeruginosa* (64%), *Klebsiella pneumoniae* (14%) and *Enterobacter spp.* (14%).

In 33% of the cases, the isolated CROs were carbapenemase producers, with a considerable majority consisting of VIM (71%), followed by KPC (22%) and OXA-48 (7%). The distribution of carbapenemases and type of infection against the isolated microorganisms are reported in Table 1. The analysis of the distribution over time showed a progressive increase in VIM production in recent years, with 100% VIM detected starting from 2021 (Figure 1).

Fifty-seven percent of the patients achieved complete and lasting clinical and microbiological recovery, whereas 5% of the cases experienced a recurrence of infection within 30 days. The mortality rate in our sample was 29%; however, only 5% of deaths were related to a CRO infection.

In our cohort, empirical antibiotic therapy was started in 52% of cases, and the most commonly used drug was a carbapenem (45%). Targeted therapy was carried out as a monotherapy in 48% of cases, and the most employed antibiotics included new-generation cephalosporins (60%); in 52% of cases, however, combination therapy was chosen, using mainly a combination of a carbapenem with Colistin (63.3%) (Table 2).

Therefore, as a second aim, we identified and compared two groups of patients within our cohort: those treated with new-generation cephalosporins and those treated with colistin-containing regimens. A statistical analysis was performed to assess whether there were significant differences in terms of outcome in the two groups of patients. The two samples were similar regarding baseline characteristics (e.g., age, gender, comorbidity, colonization, reason for CICU admission, Pediatric Mortality Index Score and Vasoactive Index Score score) (Figure 2). We compared the following outcomes in the two groups: complete recovery, death associated with CRO, all-cause 30-day-mortality, adverse reaction that caused treatment discontinuation, duration of overall therapy and targeted therapy, time of hospitalization and stay in CICU. The analysis showed that the N-CEFs-containing treatment regimen was statistically associated with complete recovery (p 0.04) and was not inferior to the COLI-containing treatment regimens in regard to the additional endpoints considered (Table 3).

## 3. Discussion

Carbapenem-resistant bacteria represent a particular concern due to the limited treatment options [11]. Most of the available literature is based on adult patient cohorts; however, the spread of carbapenemase-resistant bacteria is likewise becoming an emerging problem in the pediatric population [12]. Hence, we conducted a retrospective study in order to characterize the infections caused by CROs and compare the treatments in a cohort of vulnerable pediatric patients admitted to a cardiac ICU in our hospital in the last few years. We found that VIM carbapenemase-producing Gram-negative bacteria have increased over the years and the treatment with novel cephalosporines was safe and successful in our cohort.

In 2013, Maltezou et al. described a cohort of hospitalized pediatric patients with CRO infections. As observed in our population, pneumonia was the most frequently diagnosed infection, followed by bacteremia and urinary tract infection, and the most frequently isolated microorganism was *Pseudomonas spp*. [13]. In 2016, an Italian multicenter study of a pediatric population infected by CRE identified *Klebsiella pneumoniae* as the main isolated pathogen. In our cohort, *Klebsiella* was second after *Pseudomonas* in the list of infecting pathogens. Furthermore, the most frequently produced carbapenemase in the Italian multicenter study was KPC (67%), whereas in our study it was VIM (71%) [14]. This discrepancy could be related to the change in the epidemiology of the spread of carbapenemases over the recent years, which has seen a global increase in VIM-producing CROs [15]. Actually, in our cohort, KPC production was predominant in 2017 but was progressively replaced by VIM over the years, reaching their peak in 2021.

Moreover, in a recent study of a large cohort of patients admitted to a neonatal intensive care unit in Shanghai, 91.3% of patients with a CRO infection were previously colonized by the same organism [16]. Similarly, in our study, in 91% of cases, a match between the species of the microorganism and the type of carbapenemase produced by the colonizing organism and the etiological agent of the infection was found.

An interesting finding from our analysis was the lack of association between carbapenemase therapy in the previous 30 days and the development of an infection caused by carbapenemase-producing organisms (*p value 0.5*). This finding is consistent with a large study conducted in 2019, which showed that previous exposure to carbapenems was a risk factor for the development of non-carbapenemase-producing CRE infection. Marimuthu et al. interpreted this finding as related to a different effect of selective antibiotic pressure [17]. The available data suggest that there are different mechanisms of acquisition of resistance between non-carbapenemase-producing and carbapenemase-producing CROs: the former would acquire carbapenemase resistance through de novo mutations under selective antibiotic pressure, whereas the latter would obtain the carbapenemase production through clonal bacterial spread or horizontal gene transfer (mainly mediated by plasmids) [18,19,20].

Looking at the lack of several therapeutic options in CRO infections, the secondary aim of our study was to assess whether there were significant differences in terms of outcome in two groups of similar patients treated with N-CEFs or COLI. The low frequency of adopting new molecules to treat CRO infections is related to several factors, such as higher costs and lack of robust comparative data compared to older drugs. Older antimicrobials are still commonly used to treat CRO infections, usually in combinations [21,22]. However, the safety profiles and related limitations of these agents are well known and often have a negative impact on patient outcomes. For example, the literature suggests that colistin, alone or in combination, has no impact on prognosis and is often associated with nephrotoxicity [23]. Hence, large and robust clinical and pharmacokinetic studies are needed to compare the efficacy between ‘old’ antibiotics and new molecules. Among the new molecules, one of the most widely used antibiotics in CRO infections is Ceftazidime/avibactam, a new combination of a third-generation cephalosporin with a beta-lactamase inhibitor. Ceftazidime/avibactam was approved in the US in 2015 for the treatment of complicated abdominal and urinary infections in adults; in 2016, it was approved in Europe for the treatment of nosocomial and ventilator-associated pneumonia; and finally, in 2020, it was approved for the treatment of bacteremia [24]. Ceftolozane/tazobactam is another combination of a third-generation cephalosporin with a beta-lactamase inhibitor, approved in Europe in 2015 with the same indications as Ceftazidime/avibactam. Cefiderocol is a brand new siderophore cephalosporin, active in vitro against a variety of Ambler’s class A, C and D b-53 lactamases, and is the first agent with activity against class B b-lactamases [25].

Our statistical analysis showed that complete recovery was statistically associated with treatment with N-CEFs *(p 0.04).* This finding is in line with those reported by Van Duin in 2018 in a study comparing the use of ceftazidime/avibactam and colistin in a cohort of patients with CRE infections [26]. In 2017, Shields et al. conducted a similar study comparing ceftazidime/avibactam with a miscellaneous group of other therapeutic regimens demonstrating that the 13 patients treated with new-generation cephalosporins showed a higher therapeutic success rate [27]. Regarding the other endpoints considered in our study, no statistically significant differences between the two groups were detected. In Caston’s 2017 study of a cohort of hematological patients, treatment with Ceftazidime/avibactam was associated with a higher cure rate, a reduction in the mean duration of treatment and an equal 30-day all-cause mortality rate between the two groups [28]. In our cohort, although there were similar results for healing and 30-day all-cause mortality, treatment with N-CEFs did not appear to be associated with a reduction in the mean duration of treatment, probably because the study was conducted in a CICU where patients are often carriers of implantable devices (i.e., ventricular assistance devices and mechanical prostheses) thus therapies are often prolonged to avoid relapses. Another difference between our results and the literature data concerns the mean duration of hospitalization: in 2019, Alradaddi et al. showed a statistically significant association between Ceftazidime/avibactam treatment and reduced hospitalization [29].

Even in this case, the reason for this difference may lie in the type of population chosen for our study, characterized by a high clinical complexity often due to the very young age and the high risk of complications related to cardiac surgery in childhood.

Our data, finally, suggest that the use of new-generation cephalosporins in the pediatric population is safe: no patient presented side effects related to the administration of Ceftazidime/avibactam or the other cephalosporins. Considering these data, it can be stated that the new cephalosporins appear to be an effective and safe treatment, including pediatric patients with CRO infections.

Our study has several limitations, such as the retrospective nature of the analysis performed and the small sample size, which does not allow firm conclusions to be drawn. Therefore, considering the growing emergence of antibiotic resistance, the spread of carbapenemase-resistant bacteria and the reduced number of available therapeutic strategies, further prospective cohort studies or randomized studies are urgently needed to validate and expand the results of the present study, especially in pediatric settings.

## 4. Materials and Methods

### 4.1. Study Design and Definitions

This study was a retrospective investigation carried out from 2016 to 2022. All patients admitted to the Cardiac Intensive Care Unit of the Bambino Gesù Children's Hospital who developed an invasive infection caused by Gram-negative bacteria not susceptible to carbapenems were included. As the data in this study were collected and analyzed retrospectively, the study did not infringe upon the rights or welfare of the patients and did not require their consent.

Invasive infections were defined as isolation of a pathogen from a sterile site through culture examination of the biological specimen, that was associated with systemic symptoms. Thus, patients with sepsis, pneumonia, urinary tract infections, and abdominal and soft tissue infections were included.

Clinical cure was defined as the persistent clinical and laboratory recovery, which was associated with blood sterilization in patients with bacteremia. 

CRO-related mortality was defined as a death occurred during a CRO invasive infection not controlled by the antibiotic treatment. 

In our population, the “novel cephalosporines” used were ceftazidime/avibactam, ceftolozane/tazobactam and cefiderocol.

### 4.2. Statistical Analysis

Qualitative variables were analyzed by the Chi squared test or the Fisher exact test to assess the extent of interdependence between the different variables and the outcome. Quantitative variables were analyzed by the independent sample *t*-test. Values of *p* < 0.05 were considered statistically significant. 

### 4.3. Microbiological Cultures and Antibiotic Susceptibility Testing

After isolation and identification, all strains were characterized by antimicrobial susceptibility testing and detection of carbapenemase-encoding genes. Gram-negative bacteria obtained from blood cultures (bottles Bactec Plus aerobic/anaerobic; BD, Heidelberg, Germany) or/and biopsies, respiratory and urinary specimens, and wound swabs were identified using Matrix-Assisted Laser Desorption Ionization–Time-of-Flight Mass Spectrometry (MALDI-TOF, Bruker Daltonics, Bremen, Germany). Antimicrobial susceptibility was tested using an automated VITEK®2 system (bioMérieux, Lyon, France). The interpretation classes of bacterial susceptibility were based on the breakpoints of the European Committee on Antimicrobial Susceptibility Testing (EUCAST). The breakpoints were used to classify results into three susceptibility categories: S, susceptible standard dosing regimen; I, susceptible with increased exposure; R, resistant [30]. 

### 4.4. PCR-Based Methods for Carbapenemase Genes

Enterobacterales with a meropenem MIC (Minimum Inhibitory Concentration) >0.125 mg/L and Pseudomonas aeruginosa with multiple determinants of resistance were tested for the presence and type of carbapenemases using the molecular assay kit Xpert® Carba-R – Cepheid (Sunnyvale, California, Stati Uniti). Subsequently, the MICs were confirmed using a broth microdilution Sensititre Gram Negative DMKGN plate (ThermoFisher Scientific, Waltham, Massachusetts, USA). 

## 5. Conclusions

Our experience confirms that the spread of carbapenemase is evolving quickly in our hospital, as described worldwide. In our cohort, a significant increase in VIM carbapenemase-producing Gram-negative bacteria has been recorded in recent years. According to our knowledge, our study is the first pediatric study comparing the efficacy of new-generation cephalosporins versus colistin-containing regimens, demonstrating their promising role in patients with limited treatment options. In the future, when more treatment options are available, therapy for carbapenem-resistant bacteria should be individualized and based on molecular resistance phenotypes, susceptibility profiles, disease severity and patient characteristics.

## Figures and Tables

**Figure 1 antibiotics-12-00796-f001:**
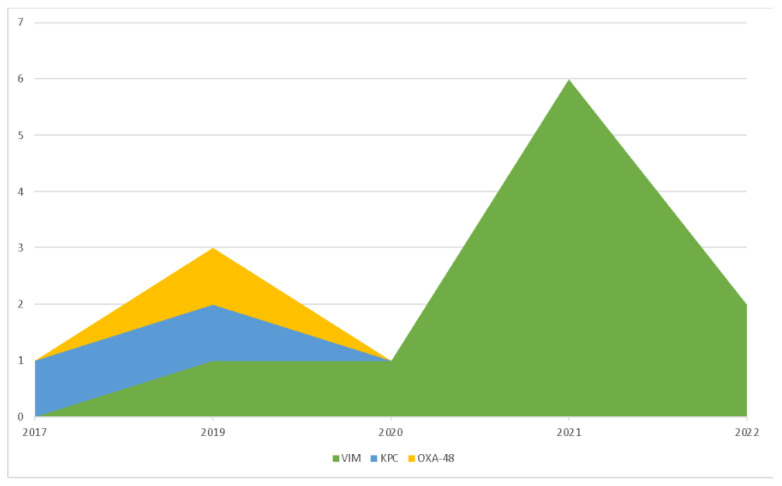
Trend of carbapenemase production during 2016–2022.

**Figure 2 antibiotics-12-00796-f002:**
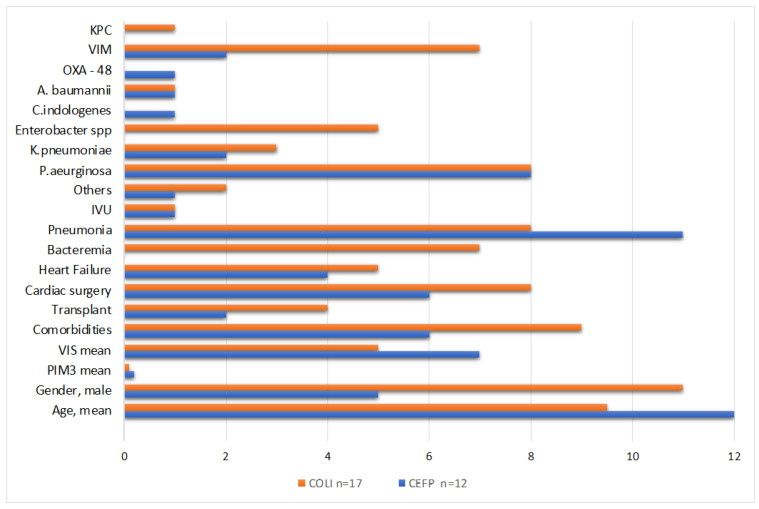
Characteristics of the two group of patients treated with new-generation cephalosporins (N-CEFs) vs. Colistin-based regimens (COLI).

**Table 1 antibiotics-12-00796-t001:** Characteristics of the infections.

	Sepsis/ Septic Shock	Pneumonia	UTI	Others	Total n° (%)
* P. aeruginosa *	6	18	2	1	**27 (64)**
Non-Carbapenemase Producers	6	15	2	1	**24 (89)**
Carbapenemase Producers	0	3	0	0	**3 (11)**
VIM	0	3	0	0	**3 (11)**
* K. pneumoniae *	3	1	1	1	**6 (14)**
Non-Carbapenemase Producers	0	0	1	1	**2 (33)**
Carbapenemase Producers	3	1	0	0	**4 (66)**
VIM	1	0	0	0	**1 (17)**
KPC	2	0	0	0	**2 (33)**
OXA 48	0	1	0	0	**1 (17)**
* Enterobacter spp. *	4	1	0	1	**6 (14)**
Non-Carbapenemase Producers	1	0	0	0	**1 (17)**
Carbapenemase Producers	3	1	0	1	**5 (83)**
VIM	3	1	0	1	**5 (83)**
* A. baumannii *	0	2	0	0	**2 (5)**
Non-Carbapenemase Producers	0	1	0	0	**1 (50)**
Carbapenemase Producers	0	1	0	0	**1 (50)**
VIM	0	1	0	0	**1 (50)**
* C. indologenes *	0	1	0	0	**1 (2)**
Non-Carbapenemase Producers	0	1	0	0	**1 (100)**

**Table 2 antibiotics-12-00796-t002:** Characteristics of the treatments.

Characteristics of the Treatment	No. (%)
**Empiric Therapy**	22 (52)
Carbapenems	10 (45)
Piperacillin/Tazobactam	2 (9)
Ceftazidime/Avibactam	6 (27)
Others (e.g., glycopetides, aminoglycosides, fluoroquinolones)	4 (20)
**Combined therapy for empiric therapy**	6 (27)
**Targeted therapy**	
**Monotherapy**	20 (48)
Ceftazidime/Avibactam	10 (50)
Colistin	2 (10)
Ceftolozane/Tazobactam	2 (10)
Cefiderocol	1 (5)
**Others (e.g., fluoroquinolones, tigecyclin)**	5 (25)
**Combined therapy**	22 (52)
Carbapenems + Colistin	14 (63.3)
Ceftazidime/Avibactam + Colistin	2 (9)
Carbapenems + Aminoglycosides	1 (4.5)
Colistin + Aminoglycosides	1 (4.5)
Others (e.g., cephalosporines + fluoroquinolones, cephalosporines + aminoglycosides)	4 (18.2)
**Adverse reactions during treatment**	0
**First line therapy failure**	3 (7)

**Table 3 antibiotics-12-00796-t003:** Outcome of patients with CRO infection treated with N-CEFs compared with patients treated with COLI-containing regimens.

	Total n = 29 (%)	N-CEFs n = 12 (%)	COLI n = 17 (%)	*p*-Value
**Clinical Cure**	13 (45)	8 (67)	5 (29)	**0.04**
Attributable mortality to CRO	1 (3)	0	1 (6)	1
30-day relapse by the same isolate	2 (7)	1 (8)	1 (6)	1
30-day all-cause mortality	4 (14)	1 (8)	3 (18)	0.62
Therapy discontinuation due to adverse events	0	0	0	
Length of overall hospitalization, median, days		179.5	116	0.46
Length of CICU stay, median, days		98	113	0.7
Overall duration of therapy, median, days		14	14	1
Duration of targeted therapy, median, days		11.5	13	0.98

## Data Availability

Data are available on reasonable request from the corresponding author.

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
