# Peer review of "The Prevalence of Carbapenemase-Producing Microorganisms and Use of Novel Cephalosporins for the Treatment of Severe Infections Caused by Carbapenem-Resistant Gram-Negative Bacteria in a Pediatric Cardiac Intensive Care Unit"

_antibiotics, 2023, doi:10.3390/antibiotics12050796_

Round 1
Reviewer 1 Report
Thank you for asking me to review this interesting paper which deals with a very important topic such as the spread of Carbapenem Resistant Organisms together with the paucity of available treatment options among the paediatric population. The findings are interesting and timely and the submission is, therefore, relevant.
I have few comments and suggestions for the Authors:
INTRO:
- - I suggest to significantly shorten the introduction. My suggestion is to limit it to 3 paragraphs as (1) state the problem, (2) what have been done/investigated so far and (3) what the Authors are planning to do with this study.
RESULTS:
- - How the CRO-related mortality was defined? I suggest to add this definition among the methods or the results.
DISCUSSION:
- - The first paragraph is a slight repetition of the intro. I suggest to replace the first paragraph with a short summary of the study findings.
METHODS:
- - I would expand the Microbiology methods a little bit (blood cultures, AST methods used)
Author Response
Thank you for reviewing our paper and, especially, for giving us some useful advice. Here, a point-by-point response.
Point 1: Intro: I suggest to significantly shorten the introduction. My suggestion is to limit it to 3 paragraphs as (1) state the problem, (2) what have been done/investigated so far and (3) what the Authors are planning to do with this study.
Response 1: As you suggested, the introduction was significantly shortened by focusing on the CRO problem, giving an overview of the current status of knowledge and explaining the aims of our study.
Point 2: Results: How the CRO-related mortality was defined? I suggest to add this definition among the methods or the results.
Response 2: As you suggested, the CRO-related mortality definition was added to the methods.
Point 3: Discussion: The first paragraph is a slight repetition of the intro. I suggest to replace the first paragraph with a short summary of the study findings
Response 3: As you proposed, the first paragraph of the discussion was shortened and a brief summary of the study findings was added.
Point 4: Methods: I would expand the Microbiology methods a little bit (blood cultures, AST methods used)
Response 3: As, you suggested, microbiology methods was expanded in the methods sections.

Reviewer 2 Report
- The method section in the abstract should be more descriptive.
- Introduction section is too long. It should be shortened to include only the relevant parts.
- Names of the bacteria must be written in italic form.
- Table 1:
o What is chistics?
o No (%) should be written as column (second column) header.
o What are “others”?
- Which new cephalosporins are used should be clearly written.
- Fig 3 is not nescessary.
- Discussion: “Klebsiella pneumoniae” instead of “Klebsiella pn”.
- Conclusion: “Our experience confirms that the spread of carbapenemase is evolving quickly and worldwide.” This is not result of this study.
Author Response
Thank you for reviewing our paper and, especially, for giving us some useful advice. Here, a point-by-point response.
Point 1: The method section in the abstract should be more descriptive.
Response 1: As you suggested, the method section was improved by adding more definitions (e.g. “CRO related-mortality”, “breakpoint”) and microbiology techniques.
Point 2: Introduction section is too long. It should be shortened to include only the relevant parts.
Response 1: As you recommended, the introduction was significantly shortened by focusing on the CRO problem, giving an overview of the current status of knowledge and explaining the aims of our study.
Point 3: Names of the bacteria must be written in italic form.
Response 1: As you recommended, all names of the bacteria were rewritten in italic form.
Point 4: Table 1:
- What is chistics?
- No (%) should be written as column (second column) header.
- What are “others”?
Response 1: I’m really sorry about “chistics”, it’s a typo. We meant “characteristics”.
“No. (%)” was written as second column header, as you suggested.
Some examples of “Others” were added in brackets in the table.
Point 5: Which new cephalosporins are used should be clearly written.
Response 1: The new cephalosporins used in the study were added in the methods section.
Point 6: Fig 3 is not necessary.
Response 6: As you suggested, the figure 3 was deleted.
Point 7: Discussion: “Klebsiella pneumoniae” instead of “Klebsiella pn”.
Response 1: “Klebsiella pn” was corrected in “Klebsiella pneumoniae”.
Point 8: Conclusion: “Our experience confirms that the spread of carbapenemase is evolving quickly and worldwide.” This is not result of this study.
Response 1: Thank you for the comment. The sentence was rewritten in order to formulate our conclusion more properly.

Reviewer 3 Report
The aim, is to review the article entitle “Trends of Carbapenemase production and use of novel Cephalosporins for the treatment of severe infections caused by carbapenem resistant organisms in a pediatric cardiac ICU”
-
I recommend to the author to rephrase the title:
-
“Trends” is not explicit enough,
-
I recommend “The incidence” or “the prevalence” of “Carbapenemase producing microorganism …”
-
I prefer “microorganism or Gram-negative” rather than “organism”, and ICU should be written in full when use for the first time, even in the title.
In the abstract,
-
“...The global spread of carbapenem resistant organisms (CROs) is an important threat to vulnerable populations worldwide, especially in intensive care unit (ICU)...” the statement is not understandable, do the authors think AMR is only a thread to vulnerable population worldwide? By adding “especially in intensive care unit” is there a direct link between vulnerable population and ICU ?
-
The abstract should include the different microorganism isolated and their resistance profile.
-
“...Carbapenemase production was present in 31% of cases…” do you mean “Carbapenamase-producing microorganisms were isolated from 31% of patients? Then, I recommend a thorough English review of this paper.
-
The results were not adequately presented in the abstract.
-
What do you mean by “progressive increase”?
-
What was your experience? Do you mean from this study? A previous study? Published?
-
“...in pediatric patients with severe CRO infection, N-CEF is a reasonable alternative to standard therapy… what is “standard therapy”? Do you mean alternative therapy for Carbapenemase-producing microorganism infection?
Introduction
-
The first paragraph is vague and does not focus on the subject. I don’t find it necessary.
-
“...Risk factors for the infections caused by CROs include prolonged exposure to antibiotics, comorbidities, invasive procedures, medical devices and mechanical ventilation… “ Please provide references
-
…predominantly fecal colonization by carbapenem-resistant Gram-negative bacteria… please provide references
-
I recommend that the introduction should be rewritten, concise and specific.
Results
-
… 58 patients with a microbiological isolate sampled from a sterile site positive for a Gram-negative carbapenem-resistant organism were retrospectively analyzed… not clear, what is sterile site? How many cases were recruited in total? How many gram-negative versus others? How many carbapenemase-producing versus others? What is the meaning of “isolated microorganisms did not reflect a clinical infection?
-
The table 1 is not clearly presented, I recommend that all the resistance profile should be presented against the isolated microorganisms. Eg: how many P. aeruginosa (should be written in italic) were VIM? KPC? OXA? Type of infection? Other resistance?
-
Figure 1: look more as a sudden stop of testing than a progressive disappearance of KPC and OXA-48. Are other phenotype tested after 2020? No new phenotypes found? Can the authors plot a monthly data rather than yearly’s.
-
Figure 2 not readable
Material and methods
-
The title should clearly state that the study is about gram-negative bacteria
-
“The interpretation classes of bacterial susceptibility to antibiotics were based on the clinical breakpoints of the European Committee on Antimicrobial Susceptibility Testing (EUCAST)” not clear , what is clinical breakpoint?
Author Response
Thank you for reviewing our paper and, especially, for giving us some useful advice. Here, a point-by-point response.
Point 1: I recommend to the author to rephrase the title:
“Trends” is not explicit enough, I recommend “The incidence” or “the prevalence” of “Carbapenemase producing microorganism …” I prefer “microorganism or Gram-negative” rather than “organism”, and ICU should be written in full when use for the first time, even in the title.
Response 1: As you suggested, the title was rephrased more properly.
Point 2: In the abstract, “...The global spread of carbapenem resistant organisms (CROs) is an important threat to vulnerable populations worldwide, especially in intensive care unit (ICU)...” the statement is not understandable, do the authors think AMR is only a thread to vulnerable population worldwide? By adding “especially in intensive care unit” is there a direct link between vulnerable population and ICU ?
Response 2: We think that AMR is a threat to the whole world, not only to vulnerable populations. However, in particular settings, such as ICU, CRO infections could be more difficult to treat due to comorbidities, altered renal function, and invasive devices of this kind of patients . We rewrote the sentence in order to clarify this concept.
Point 3: The abstract should include the different microorganism isolated and their resistance profile. “...Carbapenemase production was present in 31% of cases…” do you mean “Carbapenamase-producing microorganisms were isolated from 31% of patients? Then, I recommend a thorough English review of this paper. The results were not adequately presented in the abstract.
Response 3: The results were re-edited in order to add more information, such as the different microorganism isolated and their resistance profile. An English review of the paper was done.
Point 4: What do you mean by “progressive increase”?
Response 4: The sentence was deleted during the re-editing of the abstract.
Point 5: What was your experience? Do you mean from this study? A previous study? Published?
Response 5: The sentence was modified and we clarified that we meant the experience of the present study.
Point 6: “...in pediatric patients with severe CRO infection, N-CEF is a reasonable alternative to standard therapy… what is “standard therapy”? Do you mean alternative therapy for Carbapenemase-producing microorganism infection?
Response 6: The sentence was deleted during the re-editing of the abstract.
Point 7: Introduction. The first paragraph is vague and does not focus on the subject. I don’t find it necessary.
Response 7: As you suggested, the introduction was significantly shortened by focusing on the CRO problem, giving an overview of the current status of knowledge and explaining the aims of our study.
Point 8: “...Risk factors for the infections caused by CROs include prolonged exposure to antibiotics, comorbidities, invasive procedures, medical devices and mechanical ventilation… “ Please provide references
Response 8: As recommended, we added the reference.
Point 9: …predominantly fecal colonization by carbapenem-resistant Gram-negative bacteria… please provide references
Response 9: The sentence was deleted during the re-editing of the abstract.
Point 10: I recommend that the introduction should be rewritten, concise and specific.
Response 10: As you recommended, the introduction was re-edited.
Point 11: Results… 58 patients with a microbiological isolate sampled from a sterile site positive for a Gram-negative carbapenem-resistant organism were retrospectively analyzed… not clear, what is sterile site? How many cases were recruited in total? How many gram-negative versus others? How many carbapenemase-producing versus others? What is the meaning of “isolated microorganisms did not reflect a clinical infection?
Response 11: “Sterile site” was defined as an area of the human body where there are normally no microorganisms (e.g. blood, BAL, urine). We clarify the concept in the methods section. Since the purpose of the study was to describe CRO infections exclusively, we didn’t investigate the total amount of patients admitted to CICU during the study period; therefore, we cannot calculate the incidence of microorganisms different from carbapenem resistant Gram-negative.
The number of carbapenemase-producing microorganisms in our sample was stated in the results (33% of the total).
The sentence “Isolated microorganisms did not reflect a clinical infection” meant that, although a CRO was isolated in a usually sterile site, the patient didn’t experience any signs or symptoms of infection. We rewrote the sentence to clarify the concept.
Point 12: The table 1 is not clearly presented, I recommend that all the resistance profile should be presented against the isolated microorganisms. Eg: how many P. aeruginosa (should be written in italic) were VIM? KPC? OXA? Type of infection? Other resistance?
Response 12: We added a figure (Figure 2) in order to make this kind of information easier to read.
Point 13: Figure 1: look more as a sudden stop of testing than a progressive disappearance of KPC and OXA-48. Are other phenotype tested after 2020? No new phenotypes found? Can the authors plot a monthly data rather than yearly’s.
Response 13: All carbapenemases (OXA, VIM, KPC, IMP, NDM) were tested in every sample during the study period. However, VIM carbapenemases were detected exclusively after 2020, with a progressive disappearance of KPC and OXA-48. We didn’t succeed in plotting the monthly data because of the limited sample group.
Point 14: Figure 2 not readable
Response 14: Figure 2 was re-edited in order to make it more readable and re-named as Figure 3.
Point 15: Material and methods. The title should clearly state that the study is about gram-negative bacteria
Response 15: As you suggested, the title was modified.
Point 16: “The interpretation classes of bacterial susceptibility to antibiotics were based on the clinical breakpoints of the European Committee on Antimicrobial Susceptibility Testing (EUCAST)” not clear , what is clinical breakpoint?
Response 16: “Breakpoint” definition was added to the methods section.

Round 2
Reviewer 3 Report
1. I am not satisfied with the point12' reply. The best is to reproduce the table in two entries
The tables and figures' title are messed up. However, I don't understand why the horizontal bar plot is mixed with patient data, bacteria counts and phenotypes, please separate them for concise interpretation.
Author Response
Thank you for your useful advice. As you suggest, the table 1 was modified in order to make it easier to interpretate.
